# Relationship between primary school healthy eating and physical activity promoting environments and children's dietary intake, physical activity and weight status: a longitudinal study in the West Midlands, UK

Elizabeth Mairenn Garden,[1] Miranda Pallan [iD],[2] Joanne Clarke [iD],[2] Tania Griffin,[3] Kiya Hurley,[4] Emma Lancashire,[2] Alice J Sitch,[2,5] Sandra Passmore,[6] Peymane Adab[2]

► Prepublication history and additional materials for this paper are available online. To view these files, please visit the journal online (http://dx.doi.org/10.1136/bmjopen-2020-040833).

For numbered affiliations see end of article.

**Correspondence to**
Dr Miranda Pallan;
m.j.pallan@bham.ac.uk

## ABSTRACT

**Objective** We aimed to examine the association between food and physical activity environments in primary schools and child anthropometric, healthy eating and physical activity measures.

**Design** Observational longitudinal study using data from a childhood obesity prevention trial.

**Setting** State primary schools in the West Midlands region, UK.

**Participants** 1392 pupils who participated in the WAVES (West Midlands ActiVe lifestyle and healthy Eating in School children) childhood obesity prevention trial (2011–2015).

**Primary and secondary outcome measures** School environment (exposure) was categorised according to questionnaire responses indicating their support for healthy eating and/or physical activity. Child outcome measures, undertaken at three time points (ages 5–6, 7–8 and 8–9 years), included body mass index z-scores, dietary intake (using a 24-hour food ticklist) and physical activity (using an Actiheart monitor over 5 days). Associations between school food and physical activity environment categories and outcomes were explored through multilevel models.

**Results** Data were available for 1304 children (94% of the study sample). At age 8–9 years, children in 10 schools with healthy eating and physical activity-supportive environments had a higher physical activity energy expenditure than those in 22 schools with less supportive healthy eating/physical activity environments (mean difference=5.3 kJ/kg body weight/24 hours; p=0.05). Children in schools with supportive physical activity environments (n=8) had a lower body mass index z-score than those in schools with less supportive healthy eating/physical activity environments (n=22; mean difference=−0.17, p=0.02). School food and physical activity promoting environments were not significantly associated with dietary outcomes.

**Conclusions** School environments that support healthy food and physical activity behaviours may positively influence physical activity and childhood obesity.

> **Strengths and limitations of this study**
>
> ► The study was conducted with a large ethnically and socioeconomically diverse sample of children over a 3-year period.
> ► Outcomes were assessed by objective measures and validated instruments using standardised procedures.
> ► The school questionnaire and scoring system used to assess school food and physical activity promoting environments may not give a true representation of the extent to which schools promote health.
> ► Associations between weight, dietary and physical activity outcomes and four categories of school food/physical activity environment were explored, but this does not fully take into account the large variation in environments seen across schools.
> ► The Child and Diet Evaluation Tool only measured dietary intake over a single day, so it may not be an accurate assessment of habitual intake.

**Trial registration number** ISRCTN97000586.

## BACKGROUND

Childhood obesity is a global public health crisis[1] affecting countries of all income levels.[2] Since 2005 the overall trend has stabilised,[3] but its prevalence remains high in England, and a substantial increase in overweight and obesity prevalence is seen across the primary school years.[4] Approximately 23% of children age 4–5 years are living with overweight and obesity, and this rises to 34% by the age of 10–11 years.[4] Overweight and obesity in childhood are associated with a range of physical, psychological and social challenges[5] and are likely to persist into adulthood.[6] They also

result in significant healthcare costs, estimated to be around £6 billion per year in the UK and projected to continue to rise in the coming years.[7] Therefore, prevention strategies targeting primary school-aged children are essential to address this significant health and economic burden.

Over the last decade, prevention strategies have shifted from focusing on individuals to systems approaches,[5 8–10] recognising the multiple interconnecting physical, economic, political, social and cultural influences. Within this context, the environments in which children are situated are critical to consider.

Schools provide a physical, social and educational environment for children and have the ability to shape physical activity (PA) and eating behaviours.[5] However, to date, our understanding of how the whole school environment influences these behaviours and impacts on childhood obesity is relatively limited. Research has mainly focused on evaluating specific intervention programmes introduced into schools to target weight-related behaviours. A recent Cochrane systematic review of childhood obesity prevention interventions reported that in the primary school-aged population there is some evidence to suggest that combined dietary and PA interventions in schools had a small positive effect on body mass index z-scores (zBMI) (mean difference between intervention and control groups of −0.05, 95% CI −0.10 to −0.01; based on a meta-analysis of 20 randomised controlled trials, but with low certainty of evidence).[11] However, there was no evidence to suggest that PA or dietary interventions on their own influenced zBMI.[11]

Although previous trials, using objective outcomes, suggest that school-based interventions have the ability to increase PA levels,[12–15] a recent systematic review found no evidence of an effect of activity interventions on increasing whole-day moderate-to-vigorous physical activity (MVPA; standardised mean difference 0.02, 95% CI −0.07 to 0.11).[16]

Two systematic reviews have explored the effectiveness of school food environment interventions and policies on dietary behaviours. These have reported some beneficial effects, such as increased vegetable consumption, but have found no evidence of effect of these interventions on total energy intake.[13 17] However, there was significant heterogeneity in the included studies in terms of differences between schools and educational systems, which makes it challenging to draw general conclusions.

In general, while useful for considering specific intervention programmes designed to target diet and/or PA, the existing evidence gives little information on how the combination of different policies, initiatives and activities in schools (which together can be termed 'the school environment') promotes PA and healthy eating (HE) and consequently impacts obesity in children. Among the few studies that have examined the relationship between school-level characteristics and obesity-related behaviours, the findings have been mixed and concluded that further research is needed.[18 19]

## Objectives

In this study, using data collected as part of a large childhood obesity prevention trial,[20] we aimed to explore the association between school environments promoting HE and/or PA and anthropometric, PA and dietary outcomes in UK primary school children. We hypothesised that school environments promoting HE or PA would positively influence the corresponding behavioural and anthropometric outcomes in children.

## METHODS

### Design

This longitudinal, observational study used data obtained from the WAVES (West Midlands ActiVe lifestyle and healthy Eating in School children) trial, a UK-based cluster randomised controlled trial, conducted between 2011 and 2015, that evaluated the clinical and cost-effectiveness of a 12-month obesity prevention intervention programme delivered in primary schools.[20] The study reported here used school-level data obtained at trial baseline and participant outcome data obtained at baseline and two subsequent time points.

### Participants

All state primary schools in the West Midlands, which included year groups 1 (age 5–6 years) to 5 (age 9–10 years) and were within a 50 km radius of the University of Birmingham were eligible for inclusion in the WAVES trial. The WAVES sampling strategy sought to over-recruit schools with a high minority ethnic population through use of a weighted random sample, as the original objective of the trial was to evaluate a childhood obesity prevention intervention in an ethnically diverse population. Schools above the 80th percentile for proportion of pupils of South Asian or black ethnicity had an increased chance of being sampled, with a 3:1 ratio. To ensure representation of a range of school characteristics, the sampling strategy was further balanced to take into account the proportion of children eligible for free school meals, school size and urban/rural location. With this sampling method we selected 200 schools, which we randomly ordered and sequentially invited to participate in the trial. To achieve the required sample size of 54 schools, we approached 148 schools; 4 did not respond and 90 declined to participate.

All year 1 pupils (age 5–6 years) in participating schools were eligible to take part (n=2462 children). Parental consent was obtained for 60% of children (n=1470), of whom 1392 were measured at baseline (57% of those eligible). Child demographic data were obtained from parent questionnaires or, if these were not available, from school records. Home postcodes were obtained for children and mapped to the Index of Multiple Deprivation 2010 scores[21] to give an indicator of socioeconomic status.

### Pupil outcome measures

Participating pupils took part in assessments undertaken by trained researchers following standardised operating

procedures. Apart from demographic data, which were only obtained at baseline, all outcome measures were obtained at three time points: baseline (age 5–6 years), follow-up 1 (age 7–8 years) and follow-up 2 (age 8–9 years).

The outcomes of interest in this study were zBMI; waist circumference z-score (zWC) (both z-scores were calculated using UK 1990 reference curves for children to account for age and sex)[22]; total energy intake (kJ/24 hours); fruit and vegetable intake (g/24 hours and portions); physical activity energy expenditure (PAEE; kJ/kg body weight/24 hours); and estimated time spent in MVPA (min/24 hours; defined as time spent above the acceleration threshold of 1.75 m/s).[23] Measurement instruments and processes are summarised in table 1.

### Assessment of the school environment

Schools participating in the WAVES trial were asked to complete a baseline questionnaire which enquired about the school's HE and PA environments. The questionnaire was developed to capture the variation in these school environments across the participating schools to assist in

**Table 1** Summary of measurements undertaken and their associated outcome variables

| Measurement | Instrument | Number of measurements | Method of assessment | Outcome variable |
|---|---|---|---|---|
| Weight (to nearest 0.1 kg) | Tanita bioimpedance monitor (Tanita SC-331S; Tanita, Tokyo, Japan). | Once. | Barefoot and in light clothing. | zBMI: BMI was calculated by dividing weight (kg) by height (m$^2$). UK 1990 growth reference charts were used to produce an age-specific and sex-specific zBMI and define weight categories.[22] Overweight is defined as ≥85th centile and obese as ≥95th centile.[22] |
| Height (to nearest 0.1 cm) | Leicester height measure. | Twice (third measure if difference >0.4 cm)*. | Barefoot and in light clothing. | |
| Child demographic data: sex, ethnicity, postcode | Parent questionnaire (or school records if parent questionnaire not completed). | Once. | Parent report or school record. | |
| Waist circumference (to nearest 0.1 cm) | Flexible, non-stretch, cloth tape measure. | Twice (third measure if difference >0.4 cm)*. | Measure at iliac crest. | zWC: calculated from WC using UK reference curves.[22] |
| Dietary intake | CADET: a validated 115-item, 24-hour food ticklist completed for seven distinct time periods.[43] The CADET tool was developed for use in children aged 3–7 years.[43] | Once (24 hours). | Trained researchers recorded all food and drinks consumed by children during one school day. On the same day, children were given a home food diary for completion by a parent/carer. The following day, trained researchers collected home food diaries and reviewed these with the children. When one or more incomplete sections were identified, the researcher tried to complete these through child dietary recall. Data were processed through the CADET nutrient analysis programme at the University of Leeds. | The food and nutrient intake data were used to calculate dietary total energy intake (kJ) and fruit and vegetable intake (g) based on the Englyst method.[44] |
| Physical activity | Actiheart (Cambridge Neurotechnology, Papworth, UK). | Once (worn continuously for 5 days, including one weekend). | Attached to the child's chest with 2 ECG electrodes in school by a trained researcher, initialised to record in 30 s epochs. | Total movement volume was summarised as average acceleration, along with its underlying movement intensity distribution, using the acceleration and heart rate signals. This was used to calculate PA energy expenditure† and estimated time spent in MVPA. |

*Where two values within ≤0.4 cm, a definitive measurement value was calculated as the average of the 2. For individuals with three values recorded, a definitive measurement value was calculated as the average of the closest pair (within ≤0.4 cm) or the average of all three readings (if there were no two closest readings, but the differences between values were ≤0.4 cm). When none of the three values were within 0.4 cm of each other, no definitive measurement value was calculated.
†Children with <24 hours of valid data over the 5-day measurement period were excluded. To ensure representation across the whole 24-hour period, for those with 24 hours of valid data, only those with a distribution of at least 6 hours in each quadrant (morning: 03:00–09:00; noon: 09:00–15:00; afternoon: 15:00–21:00; and midnight: 21:00–03:00) were included.
BMI, body mass index; CADET, Child and Diet Evaluation Tool; MVPA, moderate-to-vigorous-physical activity; PA, physical activity; WC, waist circumference; zBMI, body mass index z-score; zWC, waist circumference z-score.

interpretation of the main trial findings. Question items were initially developed by the research team and then sent to a health education adviser for review. Questions were refined following feedback from the adviser. The questionnaire was completed by the head teacher or a nominated staff representative and included tickbox questions, followed by open-ended questions to gain further details regarding policies and practices in the school.

The health promoting school framework requirements, government guidelines and prior research were used to develop a scoring system to categorise schools regarding their food/PA environment based on their questionnaire responses (online supplemental table 1).[17] [24–27] The scoring system was developed in an iterative way, by the study team, until a consensus was reached on the scoring method and weighting. The use of government publications ensured that the scoring system encompassed key items that have been determined as important for national policy relating to child health at school (eg, the provision of 2 hours or more of physical education (PE) per week).[27]

Each question contributed to the final scores. Questions with continuous response variables (such as number of active after-school clubs on offer) were categorised into three groups, high, medium and low, using tertile cut-offs with a score allocation of 2, 1 and 0, respectively. Questions with categorical responses were either ordinal or binary (yes/no) questions. Binary questions, such as schools having an HE policy, were scored as 2/0. For the ordinal variables, schools allocating more than 2 hours to PE were considered of high importance and they were grouped into schools providing this for all years, some years and no years, with a score of 2, 1 and 0, while level of support from stakeholders (strong support, support and weak support) was deemed less important and so was scored out of 1 (1, 0.67 and 0.33, respectively).

We developed separate HE and PA scores based on a sum of scores from 15 and 20 questions, with a maximum possible score of 24 and 34, respectively. The scores for HE and PA were then converted into percentages. The scores ranged from 44% to 92% for HE and from 24% to 92% for PA. Using the range of scores across all schools, we used tertile cut-offs to split HE and PA scores into three groups. Schools were then categorised according to their environment: (1) complete health focus, supporting both HE and PA (top tertile for both HE and PA scores); (2) HE focus (top tertile for HE scores only); (3) PA focus (top tertile for PA scores only); and (4) minimal health focus (not in the top tertile for either HE or PA scores).

## Data analysis

All analyses were undertaken using STATA V.15.[28] Descriptive analyses were used to describe pupil characteristics at baseline and both follow-up time points. Categorical variables were summarised as number (percentage). Continuous data were summarised as mean (SD) and median (IQR), as appropriate.

Due to the clustered nature of the data, multilevel linear regression models were used, with school included as a random effect. Primary analysis explored the relationship between the school environment category and pupil outcome measures at the three study time points separately. Additionally, a model was fitted using all time points, allowing for repeated measures and including a time covariate. Variables included in the model were baseline age, sex, ethnicity, Index of Multiple Deprivation score and school environment category. Models of follow-up time points included adjustment for baseline outcome value and the arm of the WAVES trial. The trial arm covariate refers to whether schools were allocated to intervention or control arm in the main WAVES trial. Trial arm was not included in the baseline analyses as participant outcome data at this time point were collected before trial arm allocation. Model assumptions were checked using residual plots, and the appropriate inclusion of the school random effect was confirmed through likelihood ratio tests. Covariate data were missing for less than 5% of participants and therefore imputation was not performed and a complete case analysis was used.

## Patient and public involvement

The WAVES trial research team included a health education adviser (SP) who advised on engagement with schools and school questionnaire design. Input was sought from a head teacher on methods of participant recruitment and participant information materials. The oversight committee for the trial also included a head teacher who advised on trial conduct throughout.

## RESULTS

Of the 54 schools participating in the WAVES trial, 50 (93%) returned the baseline questionnaire. It is these schools and the pupil participants attending them (n=1304, 94% of the WAVES trial sample) that were included in the current analyses.

## Pupil characteristics

The baseline characteristics of the pupils are shown in table 2. The sociodemographic characteristics of the WAVES trial participants excluded from these analyses (due to non-return of school questionnaires) were compared with the study sample and did not differ significantly by age or gender. Those excluded from the analyses were less likely to be of white British ethnicity and more likely to be of South Asian ethnicity or in the most deprived Index of Multiple Deprivation score quintile. Anthropometric, dietary and PA characteristics did not differ between those included and those excluded from the analyses.

Outcome measures are summarised for all three time points in table 3. Missing data at follow-up 1 and follow-up 2 are in part due to attrition of participants as the WAVES trial progressed. The main reasons for attrition were pupils moving away from the school or being absent on

**Table 2** Summary of pupil demographic information at baseline

| | Baseline value for those analysed in this study (n=1304) |
|---|---|
| Age in years, mean (SD); missing data (n)* | 6.3 (0.31); 61 |
| Gender, n (%) | |
| Male | 654 (50.2) |
| Female | 650 (49.9) |
| Missing data* | 0 |
| Ethnicity, n (%) | |
| White British | 656 (50.8) |
| South Asian | 333 (25.8) |
| African-Caribbean | 100 (7.8) |
| Other | 202 (15.7) |
| Missing data* (n) | 13 |
| Deprivation quintile, n (%)† | |
| 1 (most deprived) | 639 (50.0) |
| 2 | 266 (20.8) |
| 3 | 146 (11.4) |
| 4 | 119 (9.3) |
| 5 (least deprived) | 109 (8.5) |
| Missing data* (n) | 25 |
| Deprivation score, mean (SD); missing data (n)* | 33.8 (17.9); 25 |

*Missing data values are not included in the denominator.
†Index of Multiple Deprivation scores categorised into five groups using the national quintile cut-offs for England.

the day of data collection. The prevalence of overweight/ obesity in the study population was 20.8% at baseline, 26.8% at follow-up 1 (15 months) and 31.1% at follow-up 2 (30 months). The prevalence varied across schools, ranging 8%–32% at baseline, 9%–47% at follow-up 1 and 9%–53% at follow-up 2. There was a statistically significant increase between baseline and follow-up 2 for zBMI and zWC (p<0.001). As would be expected, total energy intake increased over the study period as the participating children grew and increased their energy requirement. However, a statistically significant decrease in PAEE and estimated time spent in MVPA (p<0.001) was also observed during this period. There was no significant difference in fruit and vegetable intake over time (p=0.20).

### School characteristics

All schools had some activities or policies that promoted HE or PA. Overall schools gained higher scores for PA than HE promotion, with the mean normalised score (adjusted for number and type of questions) being 76% (IQR: 66%–85%) for PA and 67% (IQR: 51%–74%) for HE. Eight (16%) schools were classed as PA focus, 10 (20%) as HE focus, 10 (20%) as complete health focus

and 22 (44%) as minimal health focus. The application of the scoring system to schools based on their baseline questionnaire responses, including information on missing data, is shown in table 4.

Not all schools reported having HE or PA policies (11 (22%) with no HE policy; 3 (6%) with no PA policy); however, several of these schools were placed in the PA focus, HE focus or complete health focus categories as they scored highly in other aspects of healthy food and PA promotion activity. All schools provided drinking water throughout the school day. Thirty-five (70%) had a breakfast club available to students. However, it was difficult to assess how healthy the food provided was from the information available and only 17 schools (35%) provided a healthy break snack. Forty-five schools (94%) offered a range of activities during PE that met the Department of Education recommendations and 39 (81%) allocated two or more hours to PE per week. However, PE was taught by a specialist in only 27 schools (56%). Parents were perceived to support the HE and PA environment less than other stakeholders in the school. In addition, only 23 (46%) and 16 (33%) schools involved parents in HE and PA promotion, respectively.

### Associations between the school food/PA environment category and childhood anthropometric, diet and PA outcomes

For all models (ie, models at each time point and repeated measures models for each outcome), the regression coefficients (B) for the school environment categories are presented in table 5, with the minimal health focus category as the reference. Regression coefficients for all fixed-effect covariates in the models are presented in online supplemental tables 2–8. School environment category was not statistically significantly associated with child zBMI at baseline or follow-up 1. At follow-up 2, compared with minimal health focus schools, pupils in PA focus schools had a lower zBMI (B=−0.17, 95% CI −0.31 to −0.03, p=0.02). Although not statistically significant, compared with those in schools with minimal health focus, children in schools with a complete health focus also had a lower zBMI at follow-up 2 (B=−0.11, 95% CI −0.25 to 0.03, p=0.14). At follow-up 2, PA focus and complete health focus schools also had a lower zWC than minimal health focus schools, but this was non-significant (PA focus: B=−0.15, 95% CI −0.36 to 0.06, p=0.16; complete health focus: B=−0.18, 95% CI −0.38 to 0.03, p=0.09). Compared with minimal health focus schools, PA focus and HE focus schools had lower zBMI and zWC in the repeated measures models, but these were non-significant.

Children in schools in HE, PA and complete health focus categories had significantly lower daily PAEE at baseline than those in minimal health focus schools (PA focus: B=−6.0, 95% CI −11.7 to −0.28, p=0.04; HE focus: B=−6.3, 95% CI −11.8 to −0.86, p=0.02; and complete health focus: B=−6.3, 95% CI −11.6 to −1.0, p=0.02). They also did less MVPA at baseline; however, this was only significant for schools with a complete health focus (B=−14.1, 95% CI −27.4 to −0.75, p=0.04). By follow-up 2,

**Table 3** Summary of pupil outcome measures at three time points

| | Baseline (mean (SD) age=6.3 (0.3) years) | Follow-up 1 (mean (SD) age=7.7 (0.3) years) | Follow-up 2 (mean (SD) age=9.0 (0.3) years) |
|---|---|---|---|
| Anthropometric | | | |
| Mean (SD) zBMI; missing data* | 0.21 (1.2); 64 | 0.32 (1.3); 201 | 0.39 (1.3); 293 |
| BMI group category, n (%)† | | | |
| Underweight (≤2nd centile) | 29 (2.3) | 23 (2.1) | 23 (2.3) |
| Healthy (>2nd and <85th centiles) | 953 (76.9) | 784 (71.1) | 674 (66.7) |
| Overweight (≥85th and <95th centiles) | 109 (8.8) | 126 (11.4) | 117 (11.6) |
| Obese (≥95th centile) | 149 (12.0) | 170 (15.4) | 197 (19.5) |
| Missing data* | 64 | 201 | 293 |
| Median (IQR) zWC; missing data* | 0.76 (1.2); 167 | 0.99 (1.3); 275 | 1.1 (1.3); 416 |
| Dietary intake | | | |
| Median (IQR) dietary total energy intake (kJ/24 hours); missing data* | 6976.9 (5933.9–8101.8); 212 | 7150.3 (6100.0–8279.0); 301 | 7728.8 (6595.0–9128.9); 393 |
| Median (IQR) fruit and vegetable intake (g/24 hours); missing data* | 241.8 (149.6–352.6); 212 | 207.8 (111.5–320.2); 301 | 219.5 (117.0–346.5); 393 |
| ≥5 portion of fruits and vegetables, n (%) | | | |
| Yes | 705 (64.6) | 502 (50.1) | 519 (57.0) |
| No | 387 (35.4) | 501 (50.0) | 392 (43.0) |
| Missing data* | 212 | 301 | 393 |
| Physical activity | | | |
| Mean (SD) PAEE (kJ/kg body weight/24 hours); missing data* | 94.8 (23.8); 341 | 91.5 (24.5); 521 | 79.4 (22.4); 680 |
| Median (IQR) estimated time spent in MVPA (min/24 hours); missing data* | 58.1 (41.9–81.7); 345 | 62.0 (39.8–101.9); 523 | 43.1 (31.6–64.2); 672 |

*Missing data values are not included in the denominator.
†Based on UK 1990 reference centile curves and applying cut-offs used for population monitoring.
BMI, body mass index; MVPA, moderate-to-vigorous physical activity; PAEE, physical activity energy expenditure; zBMI, body mass index z-score; zWC, waist circumference z-score.

this trend reversed, with higher levels of both PAEE and MVPA in the HE, PA and complete health focus schools, compared with minimal health focus schools, although these findings were not significant. We found no significant differences in PAEE across school environment categories in the repeated measures model.

School environment category was not significantly associated with total dietary energy intake or fruit and vegetable intake in any of the models.

## DISCUSSION

The questionnaire responses indicated that all schools promoted healthy behaviours to some degree, but there was variation in the extent of their efforts. The school scores indicated that schools tended to focus more on promoting PA than HE. This may relate to increased government focus and spending on PA in schools during the study period.[29] Overall, we found little relationship between the school food/PA environment and pupil anthropometric measures, diet or PA levels. Most associations between the school environment category and the outcome measures were non-significant, and those that were significant were small in magnitude. However, by the second follow-up time point (30 months after baseline) there was a statistically significant lower mean zBMI among pupils in schools with PA focus, compared with those in schools with minimal health focus. Although the effect size is small (mean difference=−0.17, 95% CI −0.31 to −0.03, p=0.02), if successfully sustained it has the potential to produce public health benefits at a population level.[30] A similar finding was observed for zWC, although this was non-significant. In line with these associations between school environment activity and obesity-related outcomes, we found that at the second follow-up time point PAEE levels were higher in schools with a complete health focus compared with those with minimal health focus. Therefore, the findings may suggest that a healthy school environment could have an effect over time. In this study, we used baseline data collected when the participants had already been at school for a year, so it

**Table 4** Summary of the scoring of schools on healthy eating and physical activity based on baseline questionnaire responses

| Description of variable | Schools, n (%) | | |
|---|---|---|---|
| **Healthy eating score** | | | |
| | No (score=0) | Yes (score=2) | |
| School has a policy regarding HE* | 11 (22.5) | 38 (77.6) | |
| HE is promoted in school curriculum | 10 (20.0) | 40 (80.0) | |
| School cookery club to promote HE | 29 (58.0) | 21 (42.0) | |
| Involve parents in the promotion of HE | 27 (54.0) | 23 (46.0) | |
| Drinking water is provided throughout the day* | 0 (0.0) | 49 (100.0) | |
| Breakfast club is available to students | 15 (30.0) | 35 (70.0) | |
| Food provided by the school at break deemed as healthy* | 32 (65.3) | 17 (34.7) | |
| | Low (score=0) | Medium (score=1) | High (score=2) |
| Provide a range of ways to promote HE | 12 (24.0) | 13 (26.0) | 25 (50.0) |
| Proportion of children choosing to eat school lunches† | 12 (32.4) | 11 (29.7) | 14 (37.8) |
| | Weak support (score=0.33) | Support (score=0.67) | Strong support (score=1) |
| Governors support HE in school | 2 (4.0) | 31 (62.0) | 17 (34.0) |
| Senior leadership team support HE in school* | 0 (0.0) | 21 (42.9) | 28 (57.1) |
| Staff support HE in school* | 0 (0.0) | 27 (55.1) | 22 (44.9) |
| School council supports HE in school* | 2 (4.1) | 25 (51.0) | 22 (44.9) |
| Pupils support HE in school* | 5 (10.2) | 35 (71.4) | 9 (18.4) |
| Parents support HE in school* | 14 (28.6) | 31 (63.3) | 4 (8.2) |
| **Physical activity score** | | | |
| | No (score=0) | Yes (score=2) | |
| School has a policy regarding PA‡ | 3 (6.3) | 45 (93.8) | |
| School has a PA coordinator§ | 4 (8.5) | 43 (91.5) | |
| Staff are trained regarding PA§ | 17 (36.2) | 30 (63.8) | |
| Has sport partnerships and community links§ | 11 (23.4) | 36 (76.6) | |
| PA is promoted in the school curriculum‡ | 6 (12.5) | 42 (87.5) | |
| Involve parents in the promotion of PA‡ | 32 (66.7) | 16 (33.3) | |
| School has a walk to school campaign‡ | 13 (27.1) | 35 (72.9) | |
| PE is taught by a specialist‡ | 21 (43.8) | 27 (56.3) | |
| Range of activities offered in PE meet the Department of Education recommendations‡ | 3 (6.3) | 45 (93.8) | |
| | Low (score=0) | Medium (score=1) | High (score=2) |
| Provide a range of ways to promote PA‡ | 7 (14.6) | 20 (41.7) | 21 (43.8) |
| Variety of PA opportunities other than PE‡ | 10 (20.8) | 15 (31.3) | 23 (47.9) |
| Have a range of play equipment to be used‡ | 12 (25.0) | 18 (37.5) | 18 (37.5) |
| Provide a range of sports clubs‡ | 12 (25.0) | 11 (22.9) | 25 (52.1) |
| | For no years (score=0) | For some years (score=1) | For all years (score=2) |
| Schools allocating 2 or more hours to PE per week‡ | 7 (14.6) | 2 (4.2) | 39 (81.3) |
| | Weak support (score=0.33) | Support (score=0.67) | Strong support (score=1) |
| Governors support PA in school‡ | 0 (0.0) | 28 (58.3) | 20 (41.7) |
| Senior leadership team support PA in school‡ | 0 (0.0) | 18 (37.5) | 30 (62.5) |

Continued

| Table 4 | Continued | | |
|---|---|---|---|
| Description of variable | Schools, n (%) | | |
| Staff support PA in school‡ | 0 (0.0) | 22 (45.8) | 26 (54.2) |
| School council supports PA in school§ | 0 (0.0) | 24 (51.1) | 23 (48.9) |
| Pupils support PA in school‡ | 0 (0.0) | 22 (45.8) | 26 (54.2) |
| Parents support PA in school‡ | 4 (8.3) | 33 (68.8) | 11 (22.9) |

Missing data values are not included in the denominator used for percentage calculations.
*Missing data from 1 school.
†Missing data from 13 schools.
‡Missing data from 2 schools.
§Missing data from 3 schools.
HE, healthy eating; PA, physical activity; PE, physical education.

is possible that if we had baseline data from before school entry we would have seen a larger association between school environment and obesity and PA outcomes.

### Strengths

To our knowledge, this is the first UK study to assess the association between primary school food and PA promoting environments and child obesity measures longitudinally. The study involves a large sample, from an ethnically and socioeconomically diverse population, over a 3-year period. Outcomes were predominantly assessed by objective measures using validated instruments and standardised procedures. We accounted for clustering and individual-level confounders by using multilevel models.

### Limitations

The sample includes only 50 primary schools that consented to be part of an obesity prevention trial and parental consent was only obtained for 60% of eligible children within these schools. Our study sample may therefore represent schools with more interest in health promotion and pupils who were healthier. However, obesity prevalence in the participating schools was compared with the prevalence estimated from the National Child Measurement Programme at the time and was found to be similar.[31] Comparisons of demographic characteristics of children with and without parental consent to participate in the WAVES trial (undertaken as part of the WAVES trial) also did not show major differences,[32] suggesting that the study sample is broadly representative.

Further limitations relate to participant outcome measurement and attrition. There was attrition between each measurement time point. A higher proportion of participants of African-Caribbean ethnicity and participants from more deprived groups had missing outcome data at the follow-up time points, which may partially account for different findings in the follow-up models compared with the baseline models. Limiting analyses to participants with data at all time points would potentially help us explore this; however, we have not conducted

these analyses as they would only involve a small, unrepresentative sample.

The questionnaire used to assess the school environment was developed by the WAVES trial research team with input from a health education adviser to capture variation in the environments of the participating schools. It was assessed for face validity, but not further validated. Schools completed the questionnaire with a varying degree of completeness; therefore, the resulting information may not fully reflect their activity relating to the promotion of HE and PA. The methods for scoring schools in relation to HE and PA were developed by the authors, based on past and present guidelines relating to health promoting schools. However, the construct validity of these could not be meaningfully explored through confirmatory factor analysis due to the small number of schools in our sample.[33] Furthermore, the categorisation process of schools was pragmatically determined based on statistical cut-offs, and there may be large variations in health promoting activities within each of the categories. It is possible that, due to limitations of the baseline questionnaire and the lack of validation of the scoring system, the schools' food and PA promoting environments may not have been accurately represented, which in turn may have led to the equivocal findings of the study. In addition, the school environment was only assessed at one time point, and it is possible that support for HE or PA in the schools changed over time. Therefore, we have not been able to assess longitudinal changes in the school environment and how these are associated with obesity, diet and PA outcomes.

We explored associations between the school environment and multiple outcomes in this study. It is therefore possible that the statistically significant associations that we found could be due to multiple testing. However, as this was an exploratory, hypothesis-generating study, we did not apply a correction for multiple testing as this tends to be conservative and may have led us to miss potential differences between groups that warrant further research.[34] The Child And Diet Evaluation Tool was quick and practical with a relatively low response burden,[35] but it only measured dietary intake over a single day, so it may not be an accurate assessment of habitual intake.

**Table 5** Association between school food/physical activity environment category and pupil obesity, dietary and physical activity outcome measures, estimated using multilevel modelling

| Outcome variable | School environment category* | Outcome time point | | | | | | | |
|---|---|---|---|---|---|---|---|---|---|
| | | Baseline | | Follow-up 1 | | Follow-up 2 | | Repeated measures model | |
| | | Regression coefficient (95% CI) | P value | Regression coefficient (95% CI) | P value | Regression coefficient (95% CI) | P value | Regression coefficient (95% CI) | P value |
| zBMI | PA focus | −0.06 (−0.25 to 0.14) | 0.57 | −0.06 (−0.22 to 0.10) | 0.48 | **−0.17 (−0.31 to −0.03)** | **0.02** | −0.10 (−0.29 to 0.10) | 0.32 |
| | HE focus | −0.09 (−0.28 to 0.09) | 0.33 | −0.02 (−0.18 to 0.15) | 0.86 | −0.02 (−0.16 to 0.13) | 0.80 | −0.07 (−0.27 to 0.13) | 0.47 |
| | Complete health focus | 0.04 (−0.14 to 0.23) | 0.64 | 0.07 (−0.09 to 0.23) | 0.41 | −0.11 (−0.25 to 0.03) | 0.14 | 0.07 (−0.12 to 0.26) | 0.49 |
| zWC | PA focus | −0.10 (−0.36 to 0.16) | 0.44 | 0.04 (−0.22 to 0.30) | 0.75 | −0.15 (−0.36 to 0.06) | 0.16 | −0.11 (−0.32 to 0.11) | 0.33 |
| | HE focus | −0.24 (−0.48 to 0.01) | 0.06 | 0.19 (−0.07 to 0.45) | 0.14 | 0.07 (−0.15 to 0.29) | 0.55 | −0.14 (−0.37 to 0.08) | 0.21 |
| | Complete health focus | 0.10 (−0.14 to 0.34) | 0.40 | −0.01 (−0.26 to 0.24) | 0.95 | −0.18 (−0.38 to 0.03) | 0.09 | 0.07 (−0.14 to 0.28) | 0.52 |
| Dietary total energy intake (kJ/24 hours) | PA focus | −903.5 (−2402.7 to 595.6) | 0.24 | 433.7 (−70.5 to 937.9) | 0.09 | 33.2 (−614.5 to 681.0) | 0.92 | −220.9 (−815.1 to 373.4) | 0.47 |
| | HE focus | −847.8 (−2255.8 to 560.3) | 0.24 | −63.7 (−598.0 to 470.5) | 0.82 | 260.8 (−443.1 to 964.7) | 0.47 | −483.8 (−1087.4 to 119.9) | 0.12 |
| | Complete health focus | −399.3 (−1801.3 to 1002.6) | 0.58 | 32.8 (−464.7 to 530.3) | 0.90 | 171.3 (−466.4 to 809.1) | 0.60 | −200.0 (−773.9 to 373.8) | 0.49 |
| Fruit and vegetable intake (g/24 hours) | PA focus | −57.8 (−135.5 to 19.8) | 0.14 | 11.8 (−27.9 to 51.5) | 0.56 | −16.0 (−73.9 to 41.9) | 0.59 | −22.6 (−64.9 to 19.6) | 0.29 |
| | HE focus | −31.6 (−104.7 to 41.4) | 0.40 | 3.0 (−38.6 to 44.7) | 0.89 | −4.4 (−66.9 to 58.0) | 0.89 | −21.6 (−64.4 to 21.2) | 0.32 |
| | Complete health focus | −7.3 (−80.0 to 65.4) | 0.84 | 27.1 (−11.9 to 66.0) | 0.17 | 8.6 (−48.1 to 65.3) | 0.77 | 5.2 (−35.6 to 45.9) | 0.80 |
| PAEE (kJ/kg body weight/24 hours) | PA focus | **−6.0 (−11.7 to −0.28)** | **0.04** | −0.79 (−7.2 to 5.7) | 0.81 | 1.9 (−3.8 to 7.6) | 0.51 | −2.8 (−7.2 to 1.6) | 0.21 |
| | HE focus | **−6.3 (−11.8 to −0.86)** | **0.02** | 3.8 (−3.1 to 10.7) | 0.28 | 3.5 (−2.7 to 9.7) | 0.27 | −0.83 (−5.3 to 3.7) | 0.72 |
| | Complete health focus | **−6.3 (−11.6 to −1.0)** | **0.02** | −2.6 (−8.8 to 3.6) | 0.42 | **5.3 (0.04 to 10.6)** | **0.05** | −3.0 (−7.2 to 1.2) | 0.16 |
| Estimated time spent in MVPA (min/24 hours) | PA focus | −4.2 (−18.5 to 10.1) | 0.57 | −3.7 (−22.7 to 15.2) | 0.70 | 3.8 (−10.5 to 18.2) | 0.60 | −1.4 (−11.9 to 9.1) | 0.80 |
| | HE focus | −13.2 (−26.8 to 0.47) | 0.06 | −0.80 (−20.9 to 19.3) | 0.94 | 0.78 (−14.5 to 16.1) | 0.92 | −5.0 (−15.7 to 5.7) | 0.36 |
| | Complete health focus | **−14.1 (−27.4 to −0.75)** | **0.04** | −6.4 (−24.7 to 11.8) | 0.49 | 7.2 (−6.4 to 20.9) | 0.30 | −6.4 (−16.5 to 3.7) | 0.21 |

All models with outcomes measured at a single time point include school as a random effect, and age at baseline, sex, ethnicity and Index of Multiple Deprivation score as fixed-effect covariates. Models with outcomes at follow-up 1 and follow-up 2 also include the outcome variable at baseline and the WAVES trial arm as fixed-effect covariates.
Repeated measures models include school and pupil as a random effect, and measurement time point, age at baseline, sex, ethnicity, Index of Multiple Deprivation score and WAVES trial arm as fixed-effect covariates.
Results shown in bold are statistically significant.
*Minimal health focus schools are the reference category.
HE, healthy eating; MVPA, moderate-to-vigorous physical activity; PA, physical activity; PAEE, physical activity energy expenditure; WAVES, West Midlands ActiVe lifestyle and healthy Eating in School children; zBMI, body mass index z-score; zWC, waist circumference z-score.

Finally, the data used in this study were drawn from a randomised controlled trial evaluating a childhood obesity prevention intervention and included data from both intervention and control arms. Although there was no evidence on the effectiveness of the intervention from the main trial,[20] we adjusted for the intervention/control status of participants in our analyses. However, even with these adjustments we cannot rule out the possibility of marginal intervention effects influencing the association between school food and PA environments and anthropometric, PA and dietary outcomes.

### Comparison with previous research

This study brings new information regarding the potential impact the school HE and PA environment has on children, as previous research has focused on the effectiveness of specific interventions and policies within schools.

Our findings are in line with those from two systematic reviews that found moderate evidence that school PA interventions are able to reduce BMI.[11 13] Our study included a range of school-level activities, rather than focusing on specific interventions, and the effect size observed at 30 months was similar to the summary estimate from the trials.[11]

There is a large pool of research indicating PA interventions increase PA levels in children during the school day,[12–15] although overall PA levels may not be increased.[16] We found that children in schools with an increased focus on PA were less active and engaged in less MVPA at baseline compared with minimal health focus schools, but over time the trend was reversed, suggesting that school PA promotion may increase child PA levels over time. Our findings may also indicate that PA promotion in schools only affects PA levels as children get older, when their PA levels generally decrease.

Previous systematic reviews have not found promising results regarding the impact of interventions on dietary outcomes, although there is some evidence that they are able to increase fruit and vegetable intake.[13 17] This study did not find a relationship between the school environment and child dietary outcomes. The differences between the findings of this study and previous research may be as a result of assessing overall environment, rather than specific targeted interventions. In 2018, the House of Commons (UK Parliament) stated that there was no convincing evidence the school healthy food standards had been implemented effectively.[36] The majority (n=32, 65%) of schools in this study provided a break-time snack which was deemed unhealthy, suggesting other school food provision may also be unhealthy. Furthermore, the limitations of the questionnaire in assessing school HE environment and those related to the dietary assessment method may also have contributed to differences in findings.

In terms of healthy food and PA promoting environments, we found that school perception of parental support for HE in schools was not high (low parental support perceived by over a quarter of schools). Children consistently refer to home life being more significant in changing behaviour than schools and families have been shown to be pivotal in ensuring improvements to lifestyle in primary school-aged children.[37–39] Our findings suggest that in relation to PA and HE, parent–school relationships need strengthening. Nevertheless, the complexity of promoting a healthy school environment and of including parents in such endeavours also needs to be considered.[40]

### Implications

Consistent with previous research, this study indicates that supporting PA in primary schools may have beneficial health effects in terms of increasing PA levels and reducing weight status. Although schools often prioritise academic attainment over PE and PA, increasing PE in the curriculum has not been found to have negative academic consequences and may improve school achievement.[41 42] Therefore, schools are recommended to increase their efforts to promote PA.

In this study, school environments were not as supportive of HE as expected, which is in line with UK Parliament's report stating that there was a lack of effective implementation of school food standards legislation.[36] Stronger enforcement of existing school food regulation and further regulatory measures may be needed to ensure HE is fully supported in schools.

All associations between the school food/PA environment and health/behavioural outcomes were of small magnitude, which suggests it has an important but limited role in child health. Parents are key influencers of children's health,[38 39] but in this study the majority of schools did not involve parents in their health promotion efforts. The development of school food and PA promoting environments may be more effective if schools develop better relationships with parents and involve them in their health promotion efforts.

Finally, the changes in health behaviours that we observed were at follow-up 2, once the children had been exposed to the school environment for 3 years. This may indicate that future intervention studies should have longer follow-up periods to evaluate their effectiveness.

### CONCLUSION

We found that over time a school environment supportive of PA may positively influence zBMI and that schools with a more food and PA promoting environment may contribute to increased PA levels in children over time. More research is needed to further understand the influence of the school environment, but in the mean time schools should be encouraged to provide an environment that is supportive of PA and HE. However, schools cannot undertake health promotion in isolation. They need to be considered as part of a complex system, which requires intervention at multiple points to produce a sustained reduction in child overweight and obesity.

**Author affiliations**
[1]Medical School, University of Birmingham, Birmingham, UK
[2]Institute of Applied Health Research, University of Birmingham, Birmingham, UK
[3]Department for Health, University of Bath, Bath, UK
[4]Cancer Research UK Clinical Trials Unit, University of Birmingham, Birmingham, UK
[5]NIHR Birmingham Biomedical Research Centre, University Hospitals Birmingham NHS Foundation Trust & University of Birmingham, Birmingham, UK
[6]Services for Education, Birmingham, UK

**Acknowledgements**  We would like to acknowledge the contribution of all the WAVES study trial investigators: University of Birmingham: Peymane Adab (Professor of Chronic Disease Epidemiology and Public Health and Chief Investigator), Tim Barrett (Leonard Parsons Professor of Paediatrics and Child Health), KK Cheng (Professor of Public Health and Primary Care), Jonathan J Deeks (Professor of Biostatistics), Joan L Duda (Professor of Sport and Exercise Psychology), Emma Frew (Professor in Health Economics), Karla Hemming (Professor of Biostatistics), Miranda J Pallan (Reader in Public Health and Epidemiology) and Jayne Parry (Professor of Policy and Public Health); University of Cambridge, Cambridge MRC Epidemiology Unit/Norwegian School of Sport Sciences: Ulf Ekelund (Professor in Physical Activity and Health); University of Leeds: Janet E Cade (Professor of Nutritional Epidemiology and Public Health); Loughborough University: Amanda Daley (Professor of Behavioural Medicine); University of Edinburgh: Raj Bhopal (Emeritus Professor of Public Health); University of Warwick: Paramjit Gill (Professor of General Practice); Birmingham Community Healthcare NHS Trust: Eleanor McGee (Public Health Nutrition Lead); and Birmingham Services for Education: Sandra Passmore (Education Advisor). We thank the children, school staff and parents who participated in the trial; the study team, including Behnoush Mohammadpoor Ahranjani and Emma Popo, who helped in overseeing the study measurements and data collection; the administrative team who facilitated the running of the project; the research staff who undertook the study measurements; and Robert Lancashire, who developed the trial database and oversaw data management. We also thank our collaborators at the Institute of Metabolic Science at the Medical Research Council (MRC) Epidemiology Unit (University of Cambridge) for their support and assistance, particularly Soren Brage, Kate Westgate and Stefanie Hollidge, for training of field staff on use and maintenance of Actiheart, for overseeing data preparation, and for analysis and interpretation of the physical activity data; Timo Lakka (principal investigator of the PANIC study, University of Eastern Finland) for sharing exercise data in a similarly aged cohort of children to allow translation of observed heart rate into activity energy expenditure; staff who are working, or have previously worked, at the Nutrition Epidemiology Group in Leeds, who have supported the trial team in terms of administration of the CADET food ticklist and processing of the data to enable us to undertake dietary intake analysis, in particular Neil Hancock, Cristina Cleghorn, Meagan Christian, Jayne Hutchinson, Holly Rippin and Catherine Rycroft. We also thank Sayeed Haque for his statistical advice on the analyses for this particular study.

**Contributors**  EMG designed and undertook the analysis reported in this manuscript and drafted the manuscript. MP and PA designed and oversaw the conduct of the trial from which the data are drawn, advised on the design and analysis of the reported study, and also contributed substantially to the manuscript. JC, TG, KH and EL all designed school and pupil data collection tools and undertook data collection. They also substantially contributed to the manuscript. AJS assisted in developing the analysis plan, conducted parts of the analyses and substantially contributed to the manuscript. SP oversaw the main trial from which the data are drawn, advised on the development of the school data collection tool, and substantially contributed to the manuscript, bringing a school policy perspective. All authors approved the final version of the manuscript.

**Funding**  The WAVES trial was funded by the National Institute for Health Research (NIHR) Health Technology Assessment Programme (project reference 06/85/11). The views expressed are those of the authors and not necessarily those of the NIHR or the Department of Health and Social Care.

**Competing interests**  MP, JC, TG, KH, EL, AJS, SP and PA hold grants from the UK National Institute for Health Research. PA is Chair of the National Institute for Health Research Public Health Research Funding Committee.

**Patient consent for publication**  Not required.

**Ethics approval**  The WAVES trial was approved by NHS Research Ethics Service Committee West Midlands, the Black Country (NHS REC number: 10/H1202/69) and sponsored by the University of Birmingham (sponsor number: RG_10-183). Parents gave written informed consent for children to participate and children gave verbal

assent before measurements were taken. Permission to access the WAVES study data for this study was obtained in October 2018.

**Provenance and peer review**  Not commissioned; externally peer reviewed.

**Data availability statement**  Data are available upon reasonable request. Requests for access to data from the WAVES trial should be addressed to the principal investigator at p.adab@bham.ac.uk. All proposals requesting data access will need to specify how it is planned to use the data, and all proposals will need approval of the trial coinvestigator team before data release.

**Author note**  The trial steering committee met annually and included Kelvin Jordan (chair/statistician, Keele University), Peter Whincup (subject expert, St George's, University of London), Louise Longworth (health economist, Brunel University), John Bennett (public representative, PHSE adviser) and Peymane Adab (chief investigator, University of Birmingham).

**ORCID iDs**
Miranda Pallan http://orcid.org/0000-0002-2868-4892
Joanne Clarke http://orcid.org/0000-0003-2563-5451

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
