## [Reviewer comments · BMJ Open]

ARTICLE DETAILS

TITLE (PROVISIONAL)	Relationship between primary school healthy eating and physical activity promoting environments and children's dietary intake, physical activity and weight status: a longitudinal study in the West Midlands, United Kingdom
AUTHORS	Garden, Elizabeth; Pallan, Miranda; Clarke, Joanne; Griffin, Tania; Hurley, Kiya; Lancashire, Emma; Sitch, Alice; Passmore, Sandra; Adab, Peymane

VERSION 1 – REVIEW

REVIEWER	Andrew James Williams University of St Andrews, UK
REVIEW RETURNED	26-Jun-2020

GENERAL COMMENTS	Thank you for the opportunity to review this interesting study into the school health promotion and diet, activity and weight status. The authors use data from a trial with which some of the authors were involved. The extent to which the schools in the trial promoted health eating, and/or physical activity was assessed using a questionnaire and scoring system designed for the trial. Multilevel modelling was used to explore the association between health promotion focus and diet, physical activity and anthropometric outcomes. A focus on physical activity promotion and possible health eating promotion were found to be associated with physical activity, but no significant associations with diet or anthropometrics outcomes were observed. There is a clear rationale and need for more research on the association between culture/context and weight status. However, I have a couple of significant concerns about the assessment of the health promoting environment, and the statistical methods that need to be addressed prior to publication. Assessment of health promotion focus – In the discussion it is mentioned for the first time that the questions used to assess school environment were not designed for this purpose. This should be made clear in the methods. What was the original intended use of the data collected? You need to be clear that you asked about the provision of 2 or more hours of physical education per week, without a time period this is meaningless. It is disappointing that there does not appear to have been any public or stakeholder involvement with the development of the questionnaire. Without this involvement it is possible that the meaning of questions might have been confusing or important topics might have been missed. Confirmatory factor analysis might be useful to confirm whether the scoring of questions was appropriate. The authors recognise the questionnaire as a
--

limitation of the study, but it is really crucial to the validity and value of the study, so I think more information needs to be provided on the rigor and validity of the questionnaire.

Statistical methods – The authors point out that this is currently a unique study in its longitudinal approach (as far as I am aware), but the incorporation of the three time points is not well described in the paper. Looking at the tables in the supplemental file, it looks as though each of the main explanatory variables might have been interacted with time point, but this is not clear in the paper. These models would be assuming that each data point was independent (although clustered by school), which is not true. The repeated measures might be better managed as the lowest level in the multilevel model (time point > pupil > school). The authors may have done this, but it was not clear to me. It is not clear to me what the hypothesis was regarding the impact of the school health promotion environment over time, and how this informed the modelling. The pupils have already been attending school for a couple of years when the baseline data was collected, so may already have been exposed to the health promotion environment for a significant period of time. Maybe the impact would have been more significant if you were comparing before and after joining the school? What do the additional years of data reveal, did you expect the impact on the outcomes to increase, decrease or be maintained? Health promotion focus was only measured once, what if this changed? The schools were part of a trial, and while the trial arm is modelled, there is no discussion of the impact this could have had on the findings. It would have been good to include a sentence explaining that the model assumptions had been tested, without these we don't know if there is an issue with the modelling. There are a large number of statistical hypothesis tests reported in the paper without any mention of the risk of false positives from so many tests. Not just including 50 school, but with a sample size around 1,300 the models are adjusted for a lot of variables (31). Finally, a complete case approach is taken, and there is mention of the biases this might introduce in terms of schools with missing environment data, however, there seems to be significant missing data at each follow-up point compared to baseline which might also be introducing bias. More discussion of these biases is needed. Importantly, the sections of the STROBE form asking about missing data have been left blank, there are other relevant sections of the STROBE form which are blank, suggesting that this study has not been adequately reported.

Minor issues:

- Background line 13/14 – I can understand why it has been written as ten-to-11, but I would either use ten-to-eleven or 10-to-11.
- Objectives – These could be described in more detail than simply 'explore', what were the hypotheses? The study is inanimate to cannot itself aim to do anything: 'the aims of the study were'. Is the word data our outcomes missing from line 57/58.
- Participants line 25/26 – I think the second and in this line is unnecessary as you have the list 'free school means and, school size and location'.
- Participants line 36/37 – I don't think 'Parent questionnaires and school provided information' makes sense, should it be 'schools' or 'parent and school questionnaires'.
- Although this is a common abbreviation, zBMI is never explained in the main body of the text.

	 • Assessment of the school environment lines 37-39 – Categorical responses is sufficient, not ‘categorical response variables’. • Data analysis line 11 – Stata should be referenced • Tables – I am not sure that it is necessary to capitalise so many words in the table titles. • Discussion line 11 – The sentence in this line should be referenced. • Reference 15 – It should be van Sluijs, not can Sluijs. • Supplemental file tables – I would not report 0.000, as this is limited by the number of figures reported by Stata, and therefore represents a figure <0.001, we cannot be sure that this is 0.
--	--

REVIEWER	Louise Hardy University of Sydney, Australia
REVIEW RETURNED	01-Jul-2020

GENERAL COMMENTS	bmjopen-2020-040833: Relationship between primary school healthy eating and physical activity promoting environments and children’s dietary intake, physical activity and weight status: a longitudinal study in the West Midlands, United Kingdom Thank you for the opportunity to review the above manuscript which is based on a cluster RCT intervention (WAVES) with data collected at three time points. The findings however appear to be of only those schools that had the intervention – there is no comparison with control schools. Other publications on WAVES do indicate it was an RCT so it is not clear why this analyses is on all 54 schools and not compared with control schools. The paper appears to present findings on all schools (intervention and control). I appreciate this is a complex intervention study and potentially challenging to write but I feel there needs to be much further work on the paper for readers to better understand what happened and interpret the findings. The aim of the paper is to examine the association between primary schools’ food and activity environments and children’s weight and weight related behaviours. Whilst this is an important paper it appears to be written in haste and has omitted considerable detail. The authors refer readers to other WAVE manuscripts which is not appropriate. This paper needs to stand alone. There are important omissions in the detail of the measurements. Further, the Introduction text is very similar to previous published papers on the WAVE intervention. It appears to be cut and paste in areas. There are no details on the interventions. The authors refer to the “Health promoting school environments” which I feel is too broad – health is many things (smoke free, antibullying, sexual health etc). I feel the paper would benefit by stating schools’ food and physical activity environments. Specific comments    Page/line Comment     General There are many areas that need clarification – e.g., units of measure etc. I have identified areas that require further clarification below.   3/14 Page 7 states 54 schools.   3/16 Participants needs correcting. Actual participants were (page 7) “Parental consent was obtained for 60% of children (n=1,470), of whom 1,392 were measured at baseline.”   3/27 This sentence is not clear please correct “... body mass index z-scores, (using a 24-hour food-tick-list) and ...” That is, why is 24-hr food tick list in brackets next to BMI?   3/37 Please include response rate = 94%   3/32 “The associations between school health promotion environment categories and outcomes” Do the authors mean school food and physical activity environment?   	Page/line	Comment	General	There are many areas that need clarification – e.g., units of measure etc. I have identified areas that require further clarification below.	3/14	Page 7 states 54 schools.	3/16	Participants needs correcting. Actual participants were (page 7) “Parental consent was obtained for 60% of children (n=1,470), of whom 1,392 were measured at baseline.”	3/27	This sentence is not clear please correct “... body mass index z-scores, (using a 24-hour food-tick-list) and ...” That is, why is 24-hr food tick list in brackets next to BMI?	3/37	Please include response rate = 94%	3/32	“The associations between school health promotion environment categories and outcomes” Do the authors mean school food and physical activity environment?
Page/line	Comment														
General	There are many areas that need clarification – e.g., units of measure etc. I have identified areas that require further clarification below.														
3/14	Page 7 states 54 schools.														
3/16	Participants needs correcting. Actual participants were (page 7) “Parental consent was obtained for 60% of children (n=1,470), of whom 1,392 were measured at baseline.”														
3/27	This sentence is not clear please correct “... body mass index z-scores, (using a 24-hour food-tick-list) and ...” That is, why is 24-hr food tick list in brackets next to BMI?														
3/37	Please include response rate = 94%														
3/32	“The associations between school health promotion environment categories and outcomes” Do the authors mean school food and physical activity environment?														

	3/46	Needs greater clarification “lower body mass index z-score than those (n=22) in non-supportive environments” Is this PA and nutrition or just nutrition?
	3/50	“School health promoting environments were not significantly associated with dietary outcomes” Do the authors mean just food environment or PA and food?
	5/7	“Despite a recent plateau,” please provide time frames.
	5/9	“weight prevalence” or prevalence of overweight/obesity?
	5/14-16	“Excess weight” or prevalence of overweight/obesity?
	5/20	“...prevention strategies targeting the primary school years are...” primary school years or primary school aged children?
	5/48	“obesity-related behaviours” or weight-related? That is including overweight?
	5/48	Please delete ‘comprehensive’ – because that is what Cochrane reviews are.
	6/48	Please consider this reference which is highly relevant. Peralta, LR., et al. "Influence of School-Level Socioeconomic Status on Children's Physical Activity, Fitness, and Fundamental Movement Skill Levels." Journal of School Health 89.6 (2019): 460-467.
	7/2	The authors state WAVES is a RCT this paper appears to describe findings from all schools. Table 1 in the supplement lists school activities, but not what the interventions actually involved. How many schools and children were in the intervention and control arms?
	7/23	Why were children from low SES and CALD communities over-sampled? This needs an explanation in the Introduction – i.e., that the prevalence of overweight/obesity is higher among these children, dietary and PA behaviours are sub-optimal etc.
	7/37	“Parent questionnaires and school...” Are these for parents or are parent’s proxy reporting for their child?
	7/48	This section needs greater detail. Referring readers to other manuscripts is not appropriate. This paper needs to stand alone. For example, did 5-6 year olds complete the CADET (currently it states the researchers completed it). It was this a computer assisted interview with the child? Are 5-6 year olds reliably able to recall past 24 hour food consumption? Why was waist and not waist-to-height ratio used as a proxy metric of adiposity? Why was the waist measured at the iliac crest? (Anatomically this position is the top of the hip). Usual location is either umbilicus or narrowest point between crest of iliac and lowest intercostal. Was waist used to identify overweight and obesity? If so what were the centile cut-points? PA was measured once – but when? Which data wave? Fitted where? (waist, wrist, leg?) “¶ Children with < 24 hours of valid data were excluded.” Is this overall or per day? What demographic data were collected? DOB? Ethnicity (and how was this determined)? Residential address or just postcode? Sex? Why is the measurement N/A? Surely it was measured at least once? Please add measurement units where applicable to Table 1.
	7/57	At what point did the intervention take place?
	11/4	“iv) no health focus (all other schools).” This is not clear do the authors mean lowest tertiles of PA and HE?
	13/Table 2	Which wave are these data from? Did schools change policy practices during the intervention? (or study period)
	13/28	Is PE \geq 2 hours per day or per week?

	13/55	"... health focus categories as they scored highly in other aspects of health promotion activity." What are the 'other aspects of health promotion activity'?
	14/27	The following statement is not clear "... the study sample were less likely to be of White British ethnicity, and more likely to be of South Asian ethnicity, or in the most deprived Index of Multiple Deprivation score quintile. Anthropometric, dietary and PA characteristics did not differ between the groups" Do the authors mean 'children excluded from the analyses were less likely to be... (as above) and that there were no difference between children included and excluded in terms of sex, age, anthropometry, diet and PA characteristics? I note the protocol paper states the analyses were to be by intention to treat.
	14/Table 3	I believe it would be better to present the characteristics of the children at the start of the Results section, before the school characteristics. Are the values for missing data n or %? Is White British, Caucasian? How representative are these prevalences of ethnicity to the whole populations in which the study was conducted?
	16/Table 4	Please add mean age to column headers. Why are fruit and vegetables combined? The evidence indicates children are more likely to eat fruit than vegetables – this is important for public health interventions and messaging. (There are differences in the micronutrients of fruit and vegetables – hence better to separate).
	18/Table 5	I feel it would be useful to include the actual estimates for each outcome from the no health focus schools – it would help in interpreting the beta coefficients the other categories. Also, the number of schools in each focus category. PAEE at FU#2 for complete health focus needs to be bolded.
	19/4	"The questionnaire responses indicated that all schools promoted healthy behaviours to some degree, ..." This is confusing as the results are presented with no health focus schools as the reference category.
	Sup. Tables	It is not clear what 'Trial arm means – is this intervention or control schools? Why aren't the baseline values of the outcome available?
	Tables	Please move Table titles to above (not below) the Tables. Please report values to 1 decimal place (this would help to declutter the tables)
	References	There are inconsistencies – not all journal names italicised, mix of upper lower case in titles.
	1	

VERSION 1 – AUTHOR RESPONSE

Reviewer's comment	Our response
Reviewer 1	
Assessment of health promotion focus - In the discussion it is mentioned for the first time that the questions used to assess school	In response to the reviewer's comment we have added in the following text on page 11: "The questionnaire was developed to capture

environment were not designed for this purpose. This should be made clear in the methods. What was the original intended use of the data collected?	the variation in these school environments across the participating schools to assist in interpretation of the main trial findings.” We have also changed the relevant wording in the discussion, on page 25, to: “The questionnaire used to assess the school environment was developed by the WAVES trial research team with input from a health education adviser to capture variation in the environments of the participating schools. It was assessed for face validity, but not further validated.”
You need to be clear that you asked about the provision of 2 or more hours of physical education per week, without a time period this is meaningless.	To make this clearer we have added in further text on page 11. The relevant section now reads as: “The use of government publications ensured that the scoring system encompassed key items that have been determined as important for national policy relating to child health at school (for example the provision of two hours or more of physical education (PE) per week).”
It is disappointing that there does not appear to have been any public or stakeholder involvement with the development of the questionnaire. Without this involvement it is possible that the meaning of questions might have been confusing or important topics might have been missed.	The question items that were developed by the research team were reviewed by a health education adviser who works closely with schools. The following wording has been added on page 11: “Question items were initially developed by the research team and then sent to a health education adviser for review. Questions were refined following feedback from the adviser.”
Confirmatory factor analysis might be useful to confirm whether the scoring of questions was appropriate.	We thank the reviewer for this suggestion. Although we have not done a formal sample size calculation using Monte Carlo simulation, it is very unlikely that we have a large enough sample size to conduct this analysis, given that we would have two latent variables in our structural equation model and we only have 50 schools in the sample (please see Wolf EJ, Harrington KM, Clark SL, Miller MW. Sample size requirements for structural equation models: An evaluation of power, bias, and solution propriety. Educational and psychological measurement. 2013 Dec;73:913-34.). We have added specific wording in the limitations section of the discussion on page 26, which relates to this point: “The methods for scoring schools in relation to healthy eating and PA were developed by the authors, based on past and present

	guidelines relating to health promoting schools. However, the construct validity of these could not be meaningfully explored through confirmatory factor analysis due to the small number of schools in our sample.”
The incorporation of the three time points is not well described in the paper. Looking at the tables in the supplemental file, it looks as though each of the main explanatory variables might have been interacted with time point, but this is not clear in the paper. These models would be assuming that each data point was independent (although clustered by school), which is not true. The repeated measures might be better managed as the lowest level in the multilevel model (time point > pupil > school)	We thank the reviewer for the suggested approach to modelling. We have now undertaken additional modelling for each outcome, pupil and school included as random effects, and time point included as a fixed effect. The results are shown in Table 5, which along with the results text, has been updated. We have also edited the analysis section to more clearly describe the modelling we conducted.
The pupils have already been attending school for a couple of years when the baseline data was collected, so may already have been exposed to the health promotion environment for a significant period of time. Maybe the impact would have been more significant if you were comparing before and after joining the school?	We thank the reviewer for highlighting this point. We have now added the following wording into the discussion on page 21: “Therefore, the findings may suggest that a healthy school environment could have an effect over time. In this study, we used baseline data collected when the participants had already been at school for a year, so it is possible that if we had baseline data from before school entry, we would have seen a larger association between school environment and obesity and physical activity outcomes.”
Health promotion focus was only measured once, what if this changed?	We have now highlighted this in the limitations section of the discussion on page 26 by adding the following wording: “In addition, the school environment was only assessed at one time point, and it is possible that support for healthy eating or physical activity in the schools changed over time. Therefore, we have not been able to assess longitudinal changes in the school environment and how these are associated with obesity, diet and physical activity outcomes.”
The schools were part of a trial, and while the trail arm is modelled, there is no discussion of the impact this could have had on the findings.	We have now added the following into the limitations section of the discussion on page 27: “Finally, the data used in this study were drawn from a randomised controlled trial evaluating a childhood obesity prevention intervention and included data from both intervention and control arms. Although there

	was no evidence of effectiveness of the intervention from the main trial, we adjusted for the intervention/control status of participants in our analyses. However, even with these adjustments we cannot rule out the possibility of marginal intervention effects influencing the association between school food and PA environments and anthropometric, physical activity and dietary outcomes.”
It would have been good to include a sentence explaining that the model assumptions had been tested, without these we don't know if there is an issue with the modelling.	Thank you for highlighting. We have visually rechecked the distributions of covariates and outcomes, and where approximately normal, we have summarised using mean (SD) in the tables. We have also added the following wording to the analysis section on page 13: “Model assumptions were checked using residual plots and the appropriate inclusion of the school random effect was confirmed through likelihood ratio tests.”
There are a large number of statistical hypothesis tests reported in the paper without any mention of the risk of false positives from so many tests. Not just including 50 school, but with a sample size around 1,300 the models are adjusted for a lot of variables	Thank you for highlighting this point. We have reported multiple tests in this study as we have explored multiple outcomes. However, we have not applied a correction as this study is an exploratory study. We have acknowledged the issue of multiple testing in the discussion and have added the following wording to the limitations section on page 26-27: “We explored associations between the school environment and multiple outcomes in this study. It is therefore possible that the statistically significant associations that we found could be due to multiple testing. However, as this was an exploratory, hypothesis generating study, we did not apply a correction for multiple testing as these tend to be conservative and may have led us to miss potential differences between groups that warrant further research.”
complete case approach is taken, and there is mention of the biases this might introduce in terms of schools with missing environment data, however, there seems to be significant missing data at each follow-up point compared to baseline which might also be introducing bias. More discussion of these biases is needed.	We have now added the following to the analysis section on page 13 to clarify that this is a complete case analysis: “Covariate data were missing for less than 5% of participants, therefore imputation was not performed and a complete case analysis was used.” We have more explicitly discussed participant attrition and missing follow up data in the limitations section of the discussion, page 25, by adding the following wording:

	“There was attrition between each measurement time point. A higher proportion of participants of African/Caribbean ethnicity and participants from more deprived groups had missing outcome data at the follow up time points, which may partially account for different findings in the follow up models compared with the baseline models.”
the sections of the STROBE form asking about missing data have been left blank, there are other relevant sections of the STROBE form which are blank, suggesting that this study has not been adequately reported.	We have re-completed the STROBE checklist and uploaded it with this submission.
Background line 13/14 – I can understand why it has been written as ten-to-11, but I would either use ten-to-eleven or 10-to-11.	We have changed this to: “age 10-to-11 years”
Objectives – These could be described in more detail than simply ‘explore’, what were the hypotheses? The study is inanimate to cannot itself aim to do anything: ‘the aims of the study were’. Is the word data our outcomes missing from line 57/58.	We have edited the objectives on page 5-6 to the following: “In this study, using data collected as part of a large childhood obesity prevention trial, we aimed to explore the association between school environments promoting HE and/or PA, and anthropometric, physical activity and dietary outcomes in UK primary school children. We hypothesised that school environments promoting HE or PA would positively influence the corresponding behavioural, and anthropometric outcomes in children.”
Participants line 25/26 – I think the second and in this line is unnecessary as you have the list ‘free school means and, school size and location’.	As advised, we have removed the “and”.
Participants line 36/37 – I don’t think ‘Parent questionnaires and school provided information’ makes sense, should it be ‘schools’ or ‘parent and school questionnaires’.	We have altered edited the text on page 7 to the following: “Child demographic data were obtained from parent questionnaires, or if these were not available, from school records.”
Although this is a common abbreviation, zBMI is never explained in the main body of the text.	We have edited page 7 so the following is now stated: “The outcomes of interest in this study were: Body Mass Index z-score (zBMI); waist circumference z-score (zWC) (both z-scores were calculated using UK 1990 reference curves for children to account for age and sex)...”
Assessment of the school environment lines 37-39 – Categorical responses is sufficient, not ‘categorical response variables’.	We have taken out the word “variables”, as advised.

Data analysis line 11 – Stata should be referenced	Stata is now referenced.
Tables – I am not sure that it is necessary to capitalise so many words in the table titles.	We have now changed capitalisation in table legends.
Discussion line 11 – The sentence in this line should be referenced.	We are not quite clear on which statement the reviewer is referring to here, but can add in a reference once we have some clarification.
Reference 15 – It should be van Sluijs, not can Sluijs.	Thank you, this has now been amended.
Supplemental file tables – I would not report 0.000, as this is limited by the number of figures reported by Stata, and therefore represents a figure <0.001, we cannot be sure that this is 0.	We have changed the relevant p values in the Supplementary file tables from 0.000 to <0.001.
Reviewer 2	
Thank you for the opportunity to review the above manuscript which is based on a cluster RCT intervention (WAVES) with data collected at three time points. The findings however appear to be of only those schools that had the intervention – there is no comparison with control schools. Other publications on WAVES do indicate it was an RCT so it is not clear why this analyses is on all 54 schools and not compared with control schools. The paper appears to present findings on all schools (intervention and control).	For this study we have used data from the WAVES RCT, but we are not comparing intervention and control groups (which has been done and published previously). Rather, we are using the data we collected at baseline (through a questionnaire to school senior leader teachers) on the environments of all participating schools, and outcome data that was collected from children participating in the trial at three time points. We are therefore not considering the intervention programme that was evaluated in the WAVES trial. However, we have adjusted for whether schools were in the intervention or control arm in all regression analyses, as the intervention may have influenced the participant outcomes. We have revised the study design section on page 6 to clarify that this study used data from the WAVES trial but was not evaluating the intervention. This now reads as: “This longitudinal, observational study uses data obtained from the West Midlands ActiVe lifestyle and healthy Eating in School children (WAVES) trial; a UK-based cluster randomised controlled trial, conducted between 2011 and 2015, that evaluated the clinical and cost-effectiveness of a 12-month obesity-prevention intervention programme delivered in primary schools.[19] The study reported here used school-level data obtained

	at trial baseline and participant outcome data obtained at baseline and two subsequent time points.” We have also now added into the discussion a comment about schools being included in the study from both intervention and control arms, and that this is a potential limitation, even though adjusted for in the analysis (see response to reviewer 1).
The aim of the paper is to examine the association between primary schools’ food and activity environments and children’s weight and weight related behaviours. Whilst this is an important paper it appears to be written in haste and has omitted considerable detail. The authors refer readers to other WAVE manuscripts which is not appropriate. This paper needs to stand alone.	We have removed:  1) reference to the WAVES trial protocol (originally reference number 20), and 2) the first citation of the WAVES trial report (originally reference 24), which referred the reader to the report for more detail on the outcome measurement protocols. We have now ensured adequate detail is included on the study methods (including sampling and measures) so that this paper can be read without reference to any other WAVES papers (see pages 6-8 and Table 1). We have reviewed all other citations of papers reporting the WAVES trial and have concluded that these are appropriate.
The Introduction text is very similar to previous published papers on the WAVES intervention. It appears to be cut and paste in areas. There are no details on the interventions.	We have carefully reviewed the introduction text of previous papers published on the WAVES trial and we can confirm that we have not cut and pasted any text. There are some similarities, particularly in relation to introduction to the general topic of childhood obesity, however, this is to be expected. Overall the introduction to this paper has been developed to specifically set the scene for this particular study, which relates to the influence of the broader school environment (vs. specific school-based interventions). We have not included detail on the intervention programme evaluated in the WAVES trial, as this is not directly relevant to this study. The study reported in this manuscript is an observational cohort study, which makes use of data collected in the WAVES trial. This has now been made clearer in the manuscript (see response to reviewer’s comment above).
The authors refer to the “Health promoting school environments” which I feel is too broad—health is many things (smoke	As advised, we have changed this wording to ‘food and physical activity environments’ throughout the manuscript.

free, antibullying, sexual health etc). I feel the paper would benefit by stating schools' food and physical activity environments.	
3/14 Page 7 states 54 schools.	We have amended the manuscript so it is now clear that 54 schools were recruited to the main trial (see page participants section, page 6), and it is already stated that of the 54 schools participating in the main trial 50 returned a baseline questionnaire and so were included in the analysis for this paper (see start of Results section, page 12). In the abstract we have amended the setting to: "State-primary schools in the West Midlands region, UK."
3/16 Participants needs correcting. Actual participants were (page 7) "Parental consent was obtained for 60% of children (n=1,470), of whom 1,392 were measured at baseline."	In the abstract we have amended the participants section to: "1,392 pupils who participated in the West Midlands ActiVe lifestyle and healthy Eating in School children (WAVES) childhood obesity prevention trial (2011-2015)" We have also amended the statement that the reviewer refers to in the participants section on page 7 so it now reads: "Parental consent was obtained for 60% of children (n=1,470), of whom 1,392 were measured at baseline (57% of those eligible)."
3/27 This sentence is not clear please correct "... body mass index z-scores, (using a 24-hour food-tick-list) and ..." That is, why is 24-hr food tick list in brackets next to BMI?	We have now amended this wording in the abstract to: "body mass index z-scores, dietary intake (using a 24-hour food-tick-list)"
3/37 Please include response rate = 94%	In the results section of the abstract we have amended the wording to: "Data were available for 1,304 children (94% of the study sample)."
3/32 "The associations between school health promotion environment categories and outcomes" Do the authors mean school food and physical activity environment?	We have changed the wording in the analysis section of the abstract in the way the reviewer has suggested.
3/46 Needs greater clarification "lower body mass index z-score than those (n=22) in non-supportive environments" Is this PA and nutrition or just nutrition?	To give greater clarification in the results section of the abstract, we have amended the wording to: "Children in schools with supportive physical activity environments (n=8) had a lower body mass index z-score than those in schools with less supportive healthy eating/physical activity

	environments (n=22; mean difference=-0.166, p=0.02).
3/50 "School health promoting environments were not significantly associated with dietary outcomes" Do the authors mean just food environment or PA and food?	We have now amended this in the results section of the abstract to: "School food and physical activity promoting environments were not significantly associated with dietary outcomes."
5/7 Despite a recent plateau," please provide time frames.	The text has been amended on page 4 to: "Since 2005 the overall trend has stabilised,[3] but its prevalence remains high in England"
5/9 weight prevalence" or prevalence of overweight/obesity?	We have amended the text on page 4 to: "a substantial increase in overweight and obesity prevalence"
5/14-16 "Excess weight" or prevalence of overweight/obesity?	We have substituted "excess weight" with "overweight and obesity" throughout the manuscript.
5/20 "...prevention strategies targeting the primary school years are..." primary school years or primary school aged children?	We have amended this text on page 4: "...prevention strategies targeting primary school-aged children are essential..."
5/48 "obesity-related behaviours" or weight-related? That is including overweight?	We have amended this text on page 4 to: "target weight-related behaviours."
5/48 Please delete 'comprehensive' – because that is what Cochrane reviews are.	As advised, we have deleted the word "comprehensive" on page 4.
6/48 Please consider this reference which is highly relevant. Peralta, LR., et al. "Influence of School-Level Socioeconomic Status on Children's Physical Activity, Fitness, and Fundamental Movement Skill Levels." Journal of School Health 89.6 (2019): 460-467.	We thank the reviewer for highlighting this reference and have read it with interest. Whilst this paper explores barriers and enablers to physical activity, fitness and movement skills in Australian primary and secondary schools, its focus is on the difference between high and low SES schools in terms of the barriers and enablers. Therefore, although broadly relevant, we felt that if we included a brief discussion of this reference in the background section, we would potentially lose the focus of the background section.
7/2 The authors state WAVES is a RCT this paper appears to describe findings from all schools. Table 1 in the supplement lists school activities, but not what the interventions actually involved. How many schools and children were in the intervention and control arms?	Please see our response to the reviewer's earlier comments. This is an observational study using data from a RCT. In this study we are not evaluating the WAVES intervention.
7/23 Why were children from low SES and CALD communities over-sampled? This needs an explanation in the Introduction – i.e., that the prevalence of overweight/obesity is higher among these children, dietary and PA behaviours are sub-optimal etc.	We have added wording to the participants section of the methods on page 6 so it now reads as follows: "The WAVES sampling strategy sought to over-recruit schools with a high minority ethnic population through use of a weighted random sample, as the original objective of the trial was to evaluate a

	childhood obesity prevention intervention in an ethnically diverse population.”
7/37 “Parent questionnaires and school...” Are these for parents or are parent’s proxy reporting for their child	Please see our response to a comment from reviewer 1. We have changed the wording on page 7 to clarify this point, so it now reads as: “Child demographic data were obtained from parent questionnaires, or if these were not available, from school records.”
7/48 This section needs greater detail. Referring readers to other manuscripts is not appropriate. This paper needs to stand alone. For example, did 5-6 year olds complete the CADET (currently it states the researchers completed it). Is this a computer assisted interview with the child? Are 5-6 year olds reliably able to recall past 24 hour food consumption? Why was waist and not waist-to-height ratio used as a proxy metric of adiposity? Why was the waist measured at the iliac crest? (Anatomically this position is the top of the hip). Usual location is either umbilicus or narrowest point between crest of iliac and lowest intercostal. Was waist used to identify overweight and obesity? If so what were the centile cut-points? PA was measured once – but when? Which data wave? Fitted where? (waist, wrist, leg?) “¶ Children with < 24 hours of valid data were excluded.” Is this overall or per day? What demographic data were collected? DOB? Ethnicity (and how was this determined)? Residential address or just postcode? Sex? Why is the measurement N/A? Surely it was measured at least once? Please add measurement units where applicable to Table 1.	Please see our response to the reviewer’s earlier comment. We have ensured that the methods can be understood without reference to previous WAVES papers. We have addressed the reviewer’s comments relating to CADET, waist circumference, PA measures and demographic data in Table 1. Please note that for waist circumference measurements in children, there is no agreed consensus for an optimal location of measurement. The upper border of the iliac crest (as used in this study) is recognised as a commonly used site (see https://dapa-toolkit.mrc.ac.uk/anthropometry/objective-methods/simple-measures-circumference).
7/57 At what point did the intervention take place?	Please see our response to the reviewer’s earlier comments. This is an observational study using data from a RCT, we are not reporting evaluation of the WAVES intervention.
11/4 “iv) no health focus (all other schools).” This is not clear do the authors mean lowest tertiles of PA and HE	We have now clarified how ‘no health focus’ was defined on page 12 by adding the following in brackets: “(not in the top tertile for either HE or PA scores).”

13/Table 2 Which wave are these data from? Did schools change policy practices during the intervention? (or study period)	We have altered the text on page 17 to state: “The application of the scoring system to schools based on their baseline questionnaire responses, including information on missing data, is shown in Table 4.” We have also amended Table 4 (previously labelled Table 2) legend to: “Summary of the scoring of schools on healthy eating and physical activity based on baseline questionnaire responses”.
13/28 Is PE \geq 2 hours per day or per week?	We have amended the text in Table 4 (previously labelled Table 2) to state: “Schools allocating two or more hours to PE per week” We have also amended the text on page 19 to state: “allocated two or more hours to PE per week”.
13/55 “... health focus categories as they scored highly in other aspects of health promotion activity.” What are the ‘other aspects of health promotion activity’?	This refers to schools achieving higher scores in other areas of the scoring system (i.e. other than having a healthy eating or physical activity policy). Therefore, we have amended the text on page 18 to: “several of these schools were placed in the PA focus, HE focus or complete health focus categories as they scored highly in other aspects of food and PA promotion activity.”
14/27 The following statement is not clear “... the study sample were less likely to be of White British ethnicity, and more likely to be of South Asian ethnicity, or in the most deprived Index of Multiple Deprivation score quintile. Anthropometric, dietary and PA characteristics did not differ between the groups” Do the authors mean ‘children excluded from the analyses were less likely to be... (as above) and that there were no difference between children included and excluded in terms of sex, age, anthropometry, diet and PA characteristics? I note the protocol paper states the analyses were to be by intention to treat.	We have altered the text, on page 13, to make this clearer, so it now states: “Those excluded from the analyses were less likely to be of White British ethnicity, and more likely to be of South Asian ethnicity, or in the most deprived Index of Multiple Deprivation score quintile. Anthropometric, dietary and PA characteristics did not differ between those included and those excluded from the analyses.” We did not use intention to treat analysis in this study as it is an observational design (as explained in previous responses).
14/Table 3 I believe it would be better to present the characteristics of the children at the start of the Results section, before the school characteristics. Are the values for missing data n or %? Is White British, Caucasian?	As suggested, we have now presented the child participant characteristics at the beginning of the results section. In the participant characteristics table, the missing data are not shown in brackets and are therefore numbers, and not percentages.

How representative are these prevalences of ethnicity to the whole populations in which the study was conducted?	We have added “(n)” by “Missing data” in the left-hand column to clarify this. White British is a recognised UK census category and is not the same as caucasian (although is a sub-category).
16/Table 4 Please add mean age to column headers. Why are fruit and vegetables combined? The evidence indicates children are more likely to eat fruit than vegetables – this is important for public health interventions and messaging. (There are differences in the micronutrients of fruit and vegetables – hence better to separate).	We have now added mean (SD) age at each time point to the column headings in this Table. We thank the reviewer for highlighting the differences between fruit and vegetable intake, and agree that there are differences in consumption. However, in the UK a key public health message relates to the consumption of fruit and vegetables combined (the ‘Five-a-day’ message that encourages people to eat 5 portions of fruit and vegetables per day). Therefore we have used a combined fruit and vegetable consumption outcome.
18/Table 5 I feel it would be useful to include the actual estimates for each outcome from the no health focus schools – it would help in interpreting the beta coefficients the other categories. Also, the number of schools in each focus category. PAEE at FU#2 for complete health focus needs to be bolded.	We thank the reviewer for this suggestion, however, we feel that the key information presented in this table is the associations between the school environment category and the anthropometric, diet and physical activity outcomes, which are represented by the regression coefficients. Also, we have not added the number of schools in each category as we feel that this table already presents a lot of information, and adding further would be confusing. This information is already given in the text on page 17. We have now amended so that PAEE at follow-up two for complete health focus schools is in bold type.
19/4 “The questionnaire responses indicated that all schools promoted healthy behaviours to some degree, ...” This is confusing as the results are presented with no health focus schools as the reference category.	We have altered the text to state: “All schools had some activities or policies that promoted HE or PA”. After consideration of the reviewer’s point, we have changed the term ‘no health focus’ to ‘minimal health focus’ throughout the manuscript.
Sup Tables It is not clear what ‘Trial arm means – is this intervention or control schools? Why aren’t the baseline values of the outcome available?	We have now added an explanation of the ‘trial arm’ covariate on page 13: “The trial arm covariate refers to whether schools were allocated to intervention or control arms in the main WAVES trial. Trial arm was not included in the baseline analyses as participant

	outcome data at this time point were collected before trial arm allocation.”
Tables Please move Table titles to above (not below) the Tables. Please report values to 1 decimal place (this would help to declutter the tables)	We have moved all table titles to above the tables. We have changed all tables so that numbers greater than 1 are displayed to 1 dp and numbers less than 1 are displayed to 2 dp.
References There are inconsistencies – not all journal names italicised, mix of upper lower case in titles.	We have changed the references as advised, so they are now consistent.

VERSION 2 – REVIEW

REVIEWER	Andrew James Williams University of St Andrews, United Kingdom
REVIEW RETURNED	23-Oct-2020

GENERAL COMMENTS	I would like to express my thanks to the authors for the extensive revisions they have undertaken in response to the reviewers comments. I hope that, like me, they feel this revisions have improved the reporting and clarity of the paper. Consequently, I only have a small number of minor issues that I feel need to be addressed:  1. Abstract Analyses section: The sentence can probably just begin with ‘Associations...’ 2. Page 4, lines 13-14: referring to ‘a systems approach’ makes it sound as if there is only one of these approaches, I think the ‘a’ can be deleted 3. I think the blanket replacement of words for numbers is too extreme. For example, sentences beginning with a number should still use the word (e.g. page 5 line 10, page 19 lines 1 and 4) 4. Page 5 line 18: I don’t think the word ‘research’ is necessary 5. Page 6 line 19: I know you give the age range of specific years later in the paper, but I would give the age range of year groups 1-5 here as well. 6. Page 12 line 4: I am not sure how a range of possible scores from 44-92 can also be percentages, given that this means out of 100? Was the maximum possible score 100, or 92? If the maximum possible score was 92 and a school had that score, wouldn’t this be 100%? This point needs addressing throughout the manuscript. 7. Page 25 line 3: I am not sure ‘enables more confidence’ is the best wording here. The multilevel modelling simply accounts for the lack of independence within the sample. 8. The line requiring a reference is page 24 lines 4-5 ‘This may relate to increased government focus and spending on PA in schools during the study period.’ I think a reference is needed to support your statement that there was increased focus and spending on physical activity. 9. I wonder if there might be an additional recommendation for research resulting from the findings. The changes in health related behaviours you observed were at your second follow up time point, meaning the children had been exposed to the environment
--

	for 3 years. This might suggest that evaluations of interventions need considerable follow-up periods? 10. The bias introduced by the attrition between data collection points that you describe as potentially explaining the differences observed, might be assessed by limiting the analyses to only those pupils with complete data at all three time points. I know running this analysis would be a considerable undertaking, but if you could add anything to the paper to comment on this, it would be helpful. 11. In tables 2-8 of the supplemental file, I would limit all the p-values to 2 decimal places (<0.01 rather than <0.001) just for consistency. Also in these tables although you mention the reference category for most of the relevant variables, male is not mentioned as the reference category for gender.
--	---

VERSION 2 – AUTHOR RESPONSE

Reviewer's comment	Our response
Abstract Analyses section: The sentence can probably just begin with 'Associations...'	We have removed the 'primarily' from the Analysis section of the abstract.
Page 4, lines 13-14: referring to 'a systems approach' makes it sound as if there is only one of these approaches, I think the 'a' can be deleted	We have removed the 'a' on page 4, which now reads 'systems approaches'.
I think the blanket replacement of words for numbers is too extreme. For example, sentences beginning with a number should still use the word (e.g. page 5 line 10, page 19 lines 1 and 4)	We have altered the blanket replacement of words for numbers, ensuring that sentences start with numbers written out.
Page 5 line 18: I don't think the word 'research' is necessary	The word 'research' has been deleted on page 5: 'the existing evidence gives little information on how the combination of different policies'
Page 6 line 19: I know you give the age range of specific years later in the paper, but I would give the age range of year groups 1-5 here as well.	The sentence on page 6 has been amended as follows: 'All state primary schools in the West Midlands which included Year groups 1 (age 5-to-6 years) to 5 (age 9-to-10 years)'
Page 12 line 4: I am not sure how a range of possible scores from 44-92 can also be percentages, given that this means out of 100? Was the maximum possible score 100, or 92? If the maximum possible score was 92 and a school had that score, wouldn't this be 100%? This point needs addressing throughout the manuscript.	The sentence has been changed on page 11 to better explain the scoring: 'We developed separate HE and PA scores based on a sum of scores from 15 and 20 questions, with the maximum possible score 24 and 34 respectively. The scores for HE and PA were then converted into percentages. The scores ranged from 44-92% for HE and 24-92% for PA.'

Page 25 line 3: I am not sure 'enables more confidence' is the best wording here. The multilevel modelling simply accounts for the lack of independence within the sample.	We have removed this wording and replaced this section with the statement: 'We accounted for clustering and individual level confounders by using multilevel models.'
The line requiring a reference is page 24 lines 4-5 'This may relate to increased government focus and spending on PA in schools during the study period.' I think a reference is needed to support your statement that there was increased focus and spending on physical activity.	We have added in a reference to the Government 'PE and sport premium for primary schools' scheme.
I wonder if there might be an additional recommendation for research resulting from the findings. The changes in health related behaviours you observed were at your second follow up time point, meaning the children had been exposed to the environment for 3 years. This might suggest that evaluations of interventions need considerable follow-up periods?	We have added 'The changes in health behaviours that we observed were at follow-up-2, once the children had been exposed to the school environment for three years. This may indicate that future intervention studies should have a longer follow-up periods to evaluate their effectiveness.' on page 26.
The bias introduced by the attrition between data collection points that you describe as potentially explaining the differences observed, might be assessed by limiting the analyses to only those pupils with complete data at all three time points. I know running this analysis would be a considerable undertaking, but if you could add anything to the paper to comment on this, it would be helpful.	In the limitations section on page 22, where we discuss attrition, we have added in the following comment: 'Limiting analyses to participants with data at all time points would potentially help us explore this, however, we have not conducted these analyses as they would only involve a small, unrepresentative sample.'
In tables 2-8 of the supplemental file, I would limit all the p-values to 2 decimal places (<0.01 rather than <0.001) just for consistency. Also in these tables although you mention the reference category for most of the relevant variables, male is not mentioned as the reference category for gender.	In response to reviewer's comments we have changed decimal places accordingly. In addition, '(reference male)' has been added in to the appropriate place.

VERSION 3 – REVIEW

REVIEWER	Andrew James Williams University of St Andrews, United Kingdom
REVIEW RETURNED	24-Nov-2020

GENERAL COMMENTS	Thank you again to the authors for their efforts to address my comments. Listed below are a small number of minor errors picked up while reading the paper, which do not necessitate a further peer review. Once these are addressed I am happy that the paper is suitable for publication. Thank you.  1. Table 1 – ECG is not defined as an abbreviation in the footnotes 2. In the tables and throughout the paper SD is never defined as the abbreviation for standard deviation 3. Table 2 – IQR is defined as an abbreviations despite not being used in the table 4. Page 15 line 4, the 8 can be replaced with Eight at the start of the sentence 5. Table 5 – 95% CI is not defined as the abbreviation for 95% confidence interval 6. The tables in the supplemental file do not have any footnotes to define any of the abbreviations contained within them
---

VERSION 3 – AUTHOR RESPONSE

Many thanks for passing on these final recommendations by the peer-reviewer for alteration of the manuscript. I can confirm we have made all of the following recommended changes:

1. Table 1 – ECG is not defined as an abbreviation in the footnotes - now defined in footnotes
2. In the tables and throughout the paper SD is never defined as the abbreviation for standard deviation – now defined in footnotes of tables 2 and 3
3. Table 2 – IQR is defined as an abbreviations despite not being used in the table- this has been removed
4. Page 15 line 4, the 8 can be replaced with Eight at the start of the sentence –digit has been replaced with the word
5. Table 5 – 95% CI is not defined as the abbreviation for 95% confidence interval- now defined in footnotes
6. The tables in the supplemental file do not have any footnotes to define any of the abbreviations contained within them- footnotes defining abbreviations now added to the tables in the supplementary file

Thanks and best wishes